# Solid Harmonic Wavelet Scattering: Predicting Quantum Molecular Energy from Invariant Descriptors of 3D Electronic Densities

**Michael Eickenberg**
Department of computer science
Ecole normale supérieure
PSL Research University, 75005 Paris, France
`michael.eickenberg@nsup.org`

**Georgios Exarchakis**
Department of computer science
Ecole normale supérieure
PSL Research University, 75005 Paris, France
`georgios.exarchakis@ens.fr`

**Matthew Hirn**
Department of Computational Mathematics,
Science and Engineering;
Department of Mathematics
Michigan State University
East Lansing, MI 48824, USA
`mhirn@msu.edu`

**Stéphane Mallat**
Collège de France
Ecole Normale Supérieure
PSL Research University
75005 Paris, France

## Abstract

We introduce a solid harmonic wavelet scattering representation, invariant to rigid motion and stable to deformations, for regression and classification of 2D and 3D signals. Solid harmonic wavelets are computed by multiplying solid harmonic functions with Gaussian windows dilated at different scales. Invariant scattering coefficients are obtained by cascading such wavelet transforms with the complex modulus nonlinearity. We study an application of solid harmonic scattering invariants to the estimation of quantum molecular energies, which are also invariant to rigid motion and stable with respect to deformations. A multilinear regression over scattering invariants provides close to state of the art results over small and large databases of organic molecules.

## 1 Introduction

Deep convolutional neural networks provide state of the art results over most classification and regression problems when there is enough training data. The convolutional architecture builds a representation which translates when the input is translated. It can compute invariants to translations with a global spatial pooling operator such as averaging or max pooling. A major issue is to understand if one can reduce the amount of training data, by refining the architecture or specifying network weights, from prior information on the classification or regression problem. Beyond translation invariance, such prior information can be provided by invariance over other known groups of transformations.

This paper studies the construction of generic translation and rotation invariant representations for any 2D and 3D signals, and their application. Rotation invariant representations have been developed for 2D images, for instance in [20], where a descriptor based on oriented wavelets was used to create a jointly translation and rotation-invariant representation of texture images which retained all identity information necessary for classification. These representations have not been extended to 3D because

an oriented wavelet representation in 3D requires covering the unit sphere instead of the unit circle leading to much heavier computational requirements.

Section 2 introduces a 2D or 3D rotation invariant representation calculated with a cascade of convolutions with spherical harmonic wavelets, and modulus non-linearities. Invariance to rotations results from specific properties of spherical harmonics, which leads to efficient computations. A wavelet scattering can be implemented as a deep convolutional network where all filters are predefined by the wavelet choice [13]. In that case, prior information on invariants fully specifies the network weights. Besides translation and rotation invariance, such scattering representations linearize small deformations. Invariants to small deformations are thus obtained with linear operators applied to scattering coefficients, and scattering coefficients can provide accurate regressions of functions which are stable to deformations.

Translation and rotation invariance is often encountered in physical functionals. For example energies of isolated physical systems are usually translation and rotation invariant, and are stable to small deformations. This paper concentrates on applications to computations of quantum energies of organic molecules. Computing the energy of a molecule given the charges and the relative positions of the nuclei is a fundamental topic in computational chemistry. It has considerable industrial applications, for example to test and design materials or pharmaceuticals [4]. Density functional theory is currently the most efficient numerical technique to compute approximate values of quantum energies, but it requires considerable amounts of calculations which limit the size of molecules and the number of tests. Machine learning methods have gained traction to estimate quantum molecular energies from existing quantum chemistry databases, because they require much less computation time after training.

State of the art learning approaches have been adapted to the specificities of the underlying physics. Best results on large databases are obtained with deep neural networks whose architectures are tailored to this quantum chemistry problem. Numerical experiments in Section 4 show that applying a standard multilinear regression on our generic 3D invariant solid harmonic scattering representation yields nearly state of the art results compared to all methods, including deep neural networks, and on both small and large databases.

## 2 Solid harmonic wavelet scattering

Wavelet scattering transforms have been introduced to define representations which are invariant to translations and Lipschitz continuous to deformations [12]. In two dimensions they have been extended to define rotationally invariant representations [20] but in 3D this approach requires covering the unit sphere with multiple oriented wavelets (as opposed to the unit circle in 2D), which requires too much computation. This section introduces a solid harmonic wavelet scattering transform whose rotation invariance results from symmetries of solid harmonics. In contrast to oriented wavelets, every solid harmonic wavelet can yield its own rotation invariant descriptor because it operates in a rotational frequency space.

### 2.1 Solid harmonics in 2D and 3D

Solid harmonics are solutions of the Laplace equation $\Delta f = 0$, usually expressed in spherical coordinates, where the Laplacian is the sum of unmixed second derivatives. In 2D, interpreting $\mathbb{R}^2$ as the complex plane, we find that $z \mapsto z^\ell$ is a solution for all $\ell \in \mathbb{N}$ due to its holomorphicity[1]. Expressing this solution in polar coordinates gives $(r, \varphi) \mapsto r^\ell e^{i\ell\varphi}$, revealing an $\ell$th- order polynomial in radius and a so-called *circular harmonic* with $\ell$ angular oscillations per circle.

Solving the Laplace equation in 3D spherical coordinates $(r, \vartheta, \varphi)$ gives rise to *spherical harmonics*, the eigenvectors of the Laplacian on the sphere. Imposing separability of azimuthal and elevation contributions yields the functions $Y_\ell^m(\vartheta, \varphi) = C(\ell, m) P_\ell^m(\cos \vartheta) e^{im\varphi}$, where $P_\ell^m$ is an associated Legendre polynomial and $C(\ell, m) = \sqrt{\frac{(2\ell+1)(\ell-m)!}{4\pi(\ell+m)!}}$, for $\ell \geq 0$ and $-\ell \leq m \leq \ell$. They form an orthogonal basis of $L^2$ functions on the sphere. Analogously to the 2D case, 3D solid harmonics are

then defined as

$$(r, \vartheta, \varphi) \mapsto \sqrt{\frac{4\pi}{2\ell + 1}} r^\ell Y_\ell^m(\vartheta, \varphi).$$

## 2.2 Solid harmonic wavelets

We now define *solid harmonic wavelets* in 2D and 3D. A wavelet $\psi(u)$ is a spatial filter with zero sum, which is localized around the origin in the sense that it has a fast decay along $\|u\|$. Let $\psi_j(u) = 2^{-dj}\psi(2^{-j}u)$ be a normalized dilation of $\psi$ by $2^j$ in dimension $d$. A multiscale wavelet transform of a signal $\rho(u)$ computes convolutions with these dilated wavelets at all scales $2^j$ to obtain the set of wavelet coefficients $\{\rho \star \psi_j(u)\}_{j \in \mathbb{Z}}$. They are translation covariant. Let us denote by $\hat{\rho}(\omega)$ the Fourier transform of $\rho(u)$. The Fourier transforms of these convolutions are $\hat{\rho}(\omega)\hat{\psi}(2^j\omega)$, which yields fast computational algorithms using FFTs.

A wavelet is defined from a solid harmonic by multiplying it by a Gaussian, which localizes its support. In the 2D case we obtain the following family of wavelets:

$$\psi_\ell(r, \varphi) = \frac{1}{\sqrt{(2\pi)^2}} e^{-\frac{1}{2}r^2} r^\ell e^{i\ell\varphi}.$$

For $\ell > 0$, these functions have zero integrals and are localized around the origin. In 2D frequency polar coordinates $\omega = \lambda(\cos\alpha, \sin\alpha)^T$, one can verify that the Fourier transform of this solid harmonic wavelet is very similar to itself in signal space: $\hat{\psi}_\ell(\lambda, \alpha) = (-i)^\ell e^{-\frac{1}{2}\lambda^2} \lambda^\ell e^{i\ell\alpha}$. The solid harmonic wavelet transform inherits the rotation properties of the solid harmonics.

In 2D, the rotation of a solid harmonic incurs a complex phase shift. Let $R_\gamma \in SO(2)$ be a rotation of angle $\gamma$. We first observe that

$$R_\gamma \psi_{j,\ell}(r, \varphi) = \psi_{j,\ell}(r, \varphi - \gamma) = e^{-il\gamma}\psi(r, \varphi).$$

One can derive that rotating a signal $\rho$ produces the same rotation on its wavelet convolution, multiplied by a phase factor encoding the rotational angle: $R_\gamma \rho \star \psi_{j,\ell}(u) = e^{il\gamma} R_\gamma(\rho \star \psi_{j,\ell})(u)$. If we eliminate the phase with a modulus $U[j, \ell]\rho(u) = |\rho \star \psi_{j,\ell}(u)|$ then it becomes covariant to rotations:

$$U[j, \ell] R_\gamma \rho(u) = R_\gamma U[j, \ell]\rho(u).$$

The left of Figure 1 shows the real part of 2D solid harmonic wavelets at different scales and angular frequencies.

In 3D, solid harmonics wavelet are defined by

$$\psi_{\ell,m}(r, \vartheta, \varphi) = \frac{1}{\sqrt{(2\pi)^3}} e^{-\frac{1}{2}r^2} r^\ell Y_\ell^m(\vartheta, \varphi).$$

We write $\psi_{\ell,m,j}$ its dilation by $2^j$. Let us write $\omega$ with 3D polar coordinates: $\omega = \lambda(\cos\alpha\cos\beta, \cos\alpha\sin\beta, \sin\alpha)^T$. The Fourier transform of the wavelet has the same analytical expression up to a complex factor: $\hat{\psi}_{\ell,m}(\lambda, \alpha, \beta) = (-i)^\ell e^{-\frac{1}{2}\lambda^2} \lambda^\ell Y_\ell^m(\alpha, \beta)$. The 3D covariance to rotations is more involved. The asymmetry of the azimuthal and elevation components of the spherical harmonics requires them to be treated differently. In order to obtain a rotation covariance property, it is necessary to sum the energy over all indices $m$ for a fixed $\ell$. We thus define the wavelet modulus operator of a 3D signal $\rho(u)$ by

$$U[\ell, j]\rho(u) = \left( \sum_{m=-\ell}^{\ell} |\rho \star \psi_{\ell,m,j}(u)|^2 \right)^{1/2}.$$

Using the properties of spherical harmonics, one can prove that this summation over $m$ defines a wavelet transform modulus which is covariant to 3D rotations. For a general rotation $R \in SO(3)$

$$U[j, \ell] R\rho = R U[j, \ell]\rho.$$

## 2.3 Solid harmonic scattering invariants

We showed that the wavelet modulus $U[j, \ell]\rho$ is covariant to translations and rotations in 2D and 3D. Summing these coefficients over the spatial variable $u$ thus defines a translation and rotation invariant representation. This property remains valid under pointwise transformations, e.g. if we raise the modulus coefficients to any power $q$. Since $U[j, \ell]\rho(u)$ is obtained by a wavelet scaled by $2^j$, it is a smooth function and its integral can be computed by subsampling $u$ at intervals $2^{j-\alpha}$ where $\alpha$ is an oversampling factor typically equal to 1, to avoid aliasing. First order solid harmonic scattering coefficients in 2D and 3D are defined for any $(j_1, \ell)$ and any exponent $q$ by:

$$S[j_1, \ell, q]\rho = \sum_u \left| U[j_1, \ell]\rho(2^{j_1-\alpha}u) \right|^q$$

Translating or rotating $\rho$ does not modify $S[j_1, \ell, q]\rho$. Let $J > 0$ denote the number of scales $j_1$, and $L > 0$ the number of angular oscillations $\ell$. We choose $q \in Q = \{1/2, 1, 2, 3, 4\}$ which yields $|Q|JL$ invariant coefficients.

The summation eliminates the variability of the $U[j_1, \ell]\rho(u)$ along $u$. To avoid loosing too much information, a scattering transform retransforms this function along $u$ in order to capture the lost variabilities. This is done by calculating a convolution with a second family of wavelets at different scales $2^{j_2}$ and again computing a modulus in order to obtain coefficients which remain covariant to translations and rotations. This means that $U[j_1, \ell]\rho(u)$ is retransformed by the wavelet tranform modulus operator $U[j_2, \ell]$. Clearly $U[j_2, \ell]\, U[j_1, \ell]\rho(u)$ is still covariant to translations and rotations of $\rho$, since $U[j_1, \ell]$ and $U[j_2, \ell]$ are covariant to translations and rotations.

The variable $u$ is again subsampled at intervals $2^{j_2-\alpha}$ with an oversampling factor $\alpha$ adjusted to eliminate the aliasing. Second order scattering invariants are computed by summing over the subsampled spatial variable $u$:

$$S[j_1, j_2, \ell, q]\rho = \sum_u \left| U[j_2, \ell]\, U[j_1, \ell]\rho(2^{j_2-\alpha}u) \right|^q.$$

These coefficients are computed only for $j_2 > j_1$ because one can verify [12] that the amplitude of these invariant coefficients is negligible for $j_2 \leq j_1$. The total number of computed second order invariants is thus $|Q|LJ(J-1)/2$.

In the following, we shall write $S\rho = \{S[p]\rho\}_p$ the scattering representation of $\rho$, defined by the indices $p = (j_1, \ell, q)$ and $p = (j_1, j_2, \ell, q)$. These coefficients are computed with iterated convolutions with wavelets, modulus non-linearities, and averaging. It is proved in [13] that such wavelet convolutions and non-linearities can be implemented with a deep convolutional network, whose filters depend upon the wavelets and whose depth $J$ is the maximum scale index of all wavelets $j_1 < j_2 \leq J$.

Besides translation and rotation invariance, one can prove that a scattering transform is Lipschitz continuous to deformations [12]. This means that if $\rho(u)$ is deformed by a small (in maximum gradient norm) diffeomorphism applied to $u$, then the scattering vector stays within an error radius proportional to the size of the diffeomorphism. This property is particularly important to linearly regress functions which are also stable to deformations.

# 3 Solid harmonic scattering for quantum energy regression

We study the application of solid harmonic scattering invariants to the regression of quantum molecular energies. The next section introduces the translation and rotation invariance properties of these energies.

## 3.1 Molecular regression invariances

A molecule containing $K$ atoms is entirely defined by its nuclear charges $z_k$ and its nuclear position vectors $r_k$ indexed by $k$. Denoting by $x$ the state vector of a molecule, we have

$$x = \{(r_k, z_k) \in \mathbb{R}^3 \times \mathbb{R} : k = 1, \ldots, K\}.$$

The ground-state energy of a molecule has the following invariance properties outlined in [1]:

**Invariance to permutations** Energies do not depend on the indexation order $k$ of each nuclei;

**Isometry invariance** Energies are invariance to rigid translations, rotations, and reflections of the molecule and hence of the $r_k$;

**Deformation stability** The energy is Lipschitz continuous with respect to scaling of distances between atoms.

**Multiscale interactions** The energy has a multiscale structure, with highly energetic bonds between neighboring atoms, and weaker interactions at larger distances, such as Van-der-Waals interactions.

To regress quantum energies, a machine learning representation must satisfy the same invariance and stability properties while providing a set of descriptors which is rich enough to accurately estimate the atomization energy of a diverse collection of molecules.

A rotation invariant scattering transform has been proposed to regress quantum energies of planar molecules [9]. However this approach involves too much computations in 3D because it requires to use a large number of oriented wavelets to cover the 3D spheres. The following sections explains how to regress the energies of 3D molecules from a spherical harmonic scattering.

## 3.2 Scattering transform of an electronic density

Density Functional Theory computes molecular energies by introducing an electronic density $\rho(u)$ which specifies the probability density of presence of an electron at a point $u$. Similarly, we associate to the state vector $x$ of the molecule to a naive electronic density $\rho$ which is a sum of Gaussians densities centered on each nuclei. This density incorporates no information on chemical bounds that may arise in the molecule. For $K$ atoms placed at $\{r_k\}_{k=1}^{K}$ having charges $\{z_k\}_{k=1}^{K}$, the resulting density is

$$\rho_x(r) = \sum_{k=1}^{K} c(z_k)g(r - r_k),$$

where $g$ is a Gaussian, roughly representing an electron density localized around the nucleus, and $c(z_k)$ is a vector-valued "electronic channel". It encodes different aspects of the atomic structure. We shall use three channels: the total nuclear charge $z_k$ of the atom, the valence electronic charge $v_k$ which specifies the number of electrons which can be involved in chemical bounds, and the core electronic charge $z_k - v_k$. It results that $c(z_k) = (z_k, v_k, z_k - v_k)^T$. The molecule embedding verifies

$$\int \rho_x(u)\mathrm{d}u = \sum_{k}(z_k, v_k, z_k - v_k)^T.$$

This integral gives the total number of nucleus charges and valence and core electrons. This naive density is invariant to permutations of atom indices $k$.

The density $\rho_x$ is invariant to permutations of atom indices but it is not invariant to isometries and it can not separate multiscale interactions. These missing invariances and the separation of scales into different channels are obtained by computing its scattering representation $S\rho_x$ with solid harmonic wavelets.

In Figure 1, there is an example of a 2D solid harmonic wavelet modulus $U[j, \ell]\rho_x$ for one molecule at different scales and angular frequencies.

## 3.3 Multilinear regression

Molecular energies are regressed with multilinear combinations of scattering coefficients $S\rho_x[p]$. A multilinear regression of order $r$ is defined by:

$$\tilde{E}_r(\rho_x) = b + \sum_{i}(\nu_i \prod_{j=1}^{r}(\langle S\rho_x, w_i^{(j)}\rangle + c_i^{(j)})).$$

For $r = 1$ this is a standard linear regression. For $r = 2$ this form introduces a non-linearity similar to those found in factored gated autoencoders [14]. Trilinear regressions for $r = 3$ are also used.

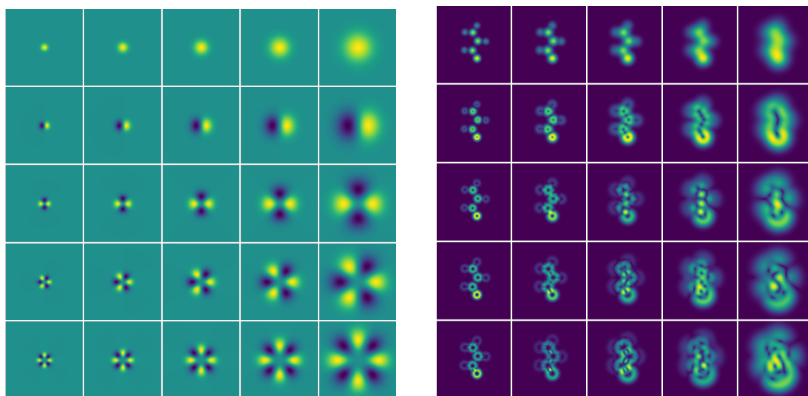

Figure 1: **Left:** Real parts of 2D solid harmonic wavelets $\psi_{\ell,j}(u)$. The $\ell$ parameters increases from $0$ to $4$ vertically where as the scale $2^j$ increases from left to right. Cartesian slices of 3D spherical harmonic wavelets yield similar patterns. **Right:** Solid harmonic wavelet moduli $S[j, \ell, 1](\rho_x)(u) = |\rho_x * \psi_{j,\ell}|(u)$ of a molecule $\rho_x$. The interference patterns at the different scales are reminiscent of molecular orbitals obtained in e.g. density functional theory.

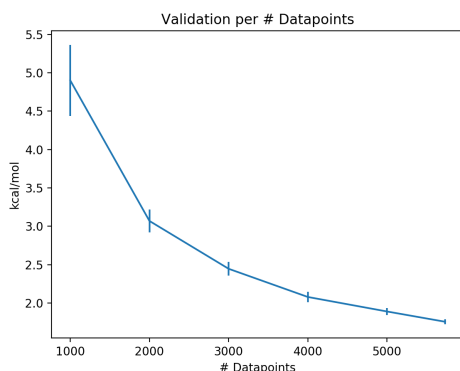

Figure 2: Mean absolute error (MAE) on the validation set as a function of the number of training points used. We observe a fast drop to low estimation errors with as few as 2000 training examples. While it is still always better to sample more of chemical space, it shows that the representation carries useful information easily amenable to further analysis, while keeping sufficient complexity to benefit from when more datapoints are available.

Here we extend the interactions to an arbitrary number of multiplicative factors. We optimize the parameters of the multilinear model by minimizing a quadratic loss function

$$L(y, \rho_x) = (y - \tilde{E}_r(\rho_x))^2$$

using the Adam algorithm for stochastic gradient descent [11]. The model described above is non-linear in the parameter space and therefore it is reasonable to assume that stochastic gradient descent will converge to a local optimum. We find that we can mitigate the effects of local optimum convergence by averaging the predictions of multiple models trained with different initializations[2].

## 4  Numerical Experiments on Chemical Databases

Quantum energy regressions are computed on two standard datasets: QM7 (GDB7-12) [18] has 7165 molecules of up to 23 atoms among H, C, O, N and S, and QM9 (GDB9-14) [17] has 133885

molecules of up to 29 atoms among H, C, O, N and F. We first review results of existing maching learning algorithms before giving results obtained with the solid harmonic scattering transform.

## 4.1  State of the art algorithms

Tables 1 and 2 gives the mean absolute error for each algorithm described below. The first machine learning approaches for quantum energy regressions were based on kernel ridge regression algorithms, optimized with different types of kernels. Kernels were first computed with Coulomb matrices, which encode pairwise nucleus-nucleus repulsion forces for each molecule [18, 15, 8, 16]. Coulomb matrices are not invariant to permutations of indices of atoms in the molecules, which leads to regression instabilities. Improvements have been obtained with bag-of-bonds descriptors [7], which groups matrix entries according to bond type, or with fixed-length smooth bond-distance histograms [2]. The BAML method (Bonds, Angles, etc, and machine learning) [10] refines the kernel by collecting atomic information, bond information, bond angle information and bond torsion information. The HDAD (Histograms of Distances, Angles, and Dihedral angles) kernels [5] improve results with computing histograms of these quantities. Smooth overlap of atomic positions (SOAP) kernels [3] can also obtain precise regression results with local descriptors computed with spherical harmonics. They are invariant to translations and rotations. However, these kernels only involve local interactions, and regression results thus degrade in presence of large-scale interactions.

Deep neural networks have also been optimized to estimate quantum molecular energies. They hold the state of the art on large databases as shown in Tables 1 and 2. Deep tensor networks [19] combine pairwise distance matrix representations in a deep learning architecture. MPNN (Message Passing Neural Networks) learns a neural network representation on the molecules represented as bond graphs. It obtains the best results on the larger QM9 data base.

## 4.2  Solid harmonic scattering results

We performed rigid affine coordinate transforms to align each molecule with its principle axis, making it possible to fit every molecule in a box of one long sidelength and two shorter ones. The Gaussian width of the electronic embedding is adjusted so that Gaussians located around the two atoms with minimal distance do not overlap too much. In all computations, the sampling grid is adjusted to keep aliasing errors negligible. Scattering vectors are standardized to have a $0$ mean and unit variance before computing the multilayer regression.

**QM7**    Scattering vectors are computed with $L = 5$. We estimated quantum energies with a linear ridge regression from scattering coefficients. The dataset comes with a split into $5$ folds, where the energy properties are approximately stratified. The average of the mean absolute error (MAE) over 5 folds is $2.4$ kcal/mol. It shows that scattering coefficients are sufficiently discriminative to obtain competitive results with a linear regression.

Bilinear regressions involve more parameters and provides near state of the art performance. We average $5$ differently initialized models over the $5$ folds to obtain a mean absolute error of $1.2$.

Figure 2 evaluates the performance of the bilinear regression on invariant scattering descriptors. From as few as 2000 training samples onward, the test set error drops below 3kcal/mol indicating that the invariant representation gives immediate access to relevant molecular properties. The fact that we observe improvement with larger data samples means that the representation also exhibits sufficient flexibility to accommodate relevant information from larger sweeps over chemical space.

**QM9**    Scattering vectors are computed with $L = 2$. Quantum energies were estimated from scattering vectors with linear, bilinear and trilinear regressions. For cross-validation, the dataset is split into $5$ folds, where the energy properties are approximately stratified. The average of the mean absolute error (MAE) over 5 folds with a trilinear regression across the 5 folds is $0.55$.

## 4.3  Discussion

The solid harmonic scattering transform followed by a multilinear regression is a domain agnostic regression scheme which only relies on prior knowledge of translation and rotation invariance as well as deformation stability. However, it leads to close to state of the art results on each data base.

| QM7 | RSCM | BoB | SOAP | DTN | CBoB | L-Scat | B-Scat |
|------|------|-----|------|-----|------|--------|--------|
| MAE | 3.1 | 1.5 | 0.9 | 1 | 1.2 | 2.4 | 1.2 |

Table 1: Mean Absolute Error in kcal/mol of quantum energy regression in QM7 for different algorithms. (RSCM: Random Sorted Coulomb Matrix[8], BoB: Bag of Bonds[7], SOAP: smooth overlap of atomic positions[3], DTN: deep tensor networks[19], CBoB: Continuous bag of bonds[2], L-Scat: Linear regression on Scattering invariants, B-Scat: Bilinear regression on Scattering invariants

| QM9 | HDAD | BAML | CM | BOB | DTN | MPNN | T-Scat |
|------|------|------|------|------|------|------|--------|
| MAE | 0.59 | 1.20 | 2.97 | 1.42 | 0.84 | 0.44 | 0.55 |

Table 2: QM9 regression results. (HDAD: Histograms of Distances, Angles and Dihedral Angles [5], BAML: Bonds, Angles and Machine Learning [10] , RSCM: Random Sorted Coulomb Matrices, BOB: Bags of Bonds, DTN: Deep Tensor Networks, MPNN: Message Passing Neural Networks [6], T-Scat: Trilinear regression on scattering invariants

The size of a scattering descriptor set grows logarithmically with the maximum number of atoms in the molecule (with increasing molecule size one continues to add scales to the wavelet transform, which adds logarithmically many coefficients) as opposed to most other methods such as [3] whose descriptor size grows linearly in the number of atoms in the molecule. Indeed, these techniques are based on measurements of local individual interactions within neighborhoods of atoms.

The representation splits the information across scales and provides scale interaction coefficients which can be related to physical phenomena as opposed to millions of deep neural net weights which are difficult to interpret. Introducing multilinear regression between solid harmonic wavelet invariants further improves the performance on the energy regression task, achieving near state of the art performance. This may also be related to multilinear expansions of physical potentials.

It is important to issue a word of caution on the chemical interpretation of these algorithmic regressions. Indeed, all data bases are computed with DFT numerical codes, which only approximate the energy. For the QM9 database, validation errors are on average 5 kcal/mol [17] on calculated energies compared to true chemical energies of ground state molecules. Refined results of fractions of kcal/mol thus no longer add true chemical information but rather reflect the ability to estimate the values produced by DFT numerical codes.

## 5 Conclusion

We introduced a 2D and 3D solid harmonic wavelet scattering transform which is invariant to translations and rotations and stable to deformations. It is computed with two successive convolutions with solid harmonic wavelets and complex modulus. Together with multilinear regressions, this representation provides near state of the art results for estimation of quantum molecular energies. The same representation is used for small and large data bases. The mathematical simplicity of these descriptors opens the possibility to relate these regression to multiscale properties of quantum chemical interactions.

**Acknowledgements**

M.E., G.E. and S.M. are supported by ERC grant InvariantClass 320959; M.H. is supported by the Alfred P. Sloan Fellowship, the DARPA YFA, and NSF grant 1620216.

## Footnotes

[1]Real and imaginary parts of holomorphic functions are harmonic - their Laplacian is 0

[2]For implementation details see `http://www.di.ens.fr/data/software/`

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
