[Reviews · NeurIPS 2017]

Reviewer 1



The paper presents the solid harmonic scattering, which creates a rotation invariant representation of 2D and 3D structures. The paper presents the details of the proposed transformation and derives its properties. The solid harmonic scattering is then used to predict the energy of molecules given the positions of individual atoms and their charges. A permutation invariant embedding of a molecule is first computed from positions and charges, and then the scattering transform is applied to obtain a rotation and translation invariance representation. This representation is used to predict the total energy with a linear regressor and a neural net with multiplicative gates. Experiments in the GDB7-12 dataset are performed and the results are competitive with other machine learning based approaches. The problem of energy prediction is important, and the proposed transformation is interesting. The introduction makes the case of learning from data using the right operators (such as convolutions for images), and motivates the exploration of special operators to analyze other types of data, such as molecule structures. The authors implemented the solution in GPUs to accelerate computations. The formulation of the method seems interesting, but the following questions need to be answered to frame this work with the current research in machine learning: * Although an elegant design, the proposed module does not have learnable parameters, and thus the contribution to machine learning appears limited. Convolutional filters have the capacity to adapt weights for each problem, while the proposed transform seems to be fixed. Even though it has connections to neural networks, the static nature of the solution goes in the opposite direction of designing machines that learn. * The transform seems to be very specific for molecule representation. It would be interesting to see applications in other domains that would benefit from this transform. If the value of this work has more impact in the quantum physics community, perhaps NIPS is the wrong venue to discuss its merit?

Reviewer 2



This paper presents a method for computing rotation and translation-invariant filters for 2D and 3D signals. Invariance is acheived by integrating over the (modulus of) solid harmonic wavelet coefficients. The utility of solid harmonic wavelets to achieve transformation invariance seems like a novel contribution to this application (e.g. molecular energy regression). However, the proposed invariant coefficients are calculated in a fixed way. The neural network is actually applied only for regression on the invariant descriptors for example. Thus, this proposed work has a much in common with the long line of literature that has explored rotation invariance for 3D signal representations using Spherical Harmonic expansions (e.g. Rotation Invariant Spherical Harmonic Representation of 3D Shape Descriptors 2003 just to list one of many). Given the close tie between the wavelet construction and spherical/solid harmonics, it would be easier to track the background of this paper by exploring the relationship to the existing literature on invariance through spherical harmonics. The contribution to the molecular energy prediction seems interesting and novel, and the impact of the paper may depend on the value of that contribution (this review is not an expert in that topic). Based on the invariant descriptors alone however, the paper is borderline (experimental impact would be necessary).

Reviewer 3



This paper presents a novel representation learning approach for 2D and 3D images, based on solid harmonic wavelet scattering. The representations are built to be invariant to rotation and translation. The proposed approach is applied to quantum molecular energy prediction and is shown to reach state-of-the-art performance. The paper is well written. The solid harmonic wavelet scattering approach is a significant contribution and the application to quantum energy regression is very interesting. The experiments are convincing and shows promising results. I am a bit concerned by the potential applications of the proposed approach: The wavelets seem to be specifically designed for the task at hand, the authors should discuss other potential applications in more details. Minor comments: - References should be added for the claims in Section 3.1 - Please add references to Table 1, either directly in the table or in the caption.